# Landscape Genetics of the Yellow-Bellied Toad (*Bombina variegata*) in the Northern Weser Hills of Germany



**Jasmin Kleißen [1], Niko Balkenhol [2] and Heike Pröhl [1],***

[1] Institute of Zoology, University of Veterinary Medicine Hannover, Bünteweg 17, 30599 Hannover, Germany; jasmin.kleissen@web.de

[2] Wildlife Sciences, University of Göttingen, Büsgenweg 3, 37077 Göttingen, Germany; niko.balkenhol@forst.uni-goettingen.de

\* Correspondence: heike.proehl@tiho-hannover.de; Tel.: +49-511-953-8431

**Abstract:** Anthropogenic influences such as deforestation, increased infrastructure, and general urbanization has led to a continuous loss in biodiversity. Amphibians are especially affected by these landscape changes. This study focuses on the population genetics of the endangered yellow-bellied toad (*Bombina variegata*) in the northern Weser Hills of Germany. Additionally, a landscape genetic analysis was conducted to evaluate the impact of eight different landscape elements on the genetic connectivity of the subpopulations in this area. Multiple individuals from 15 study sites were genotyped using 10 highly polymorphic species-specific microsatellites. Four genetic clusters were detected, with only two of them having considerable genetic exchange. The average genetic differentiation between populations was moderate (global $F_{ST} = 0.1$). The analyzed landscape elements showed significant correlations with the migration rates and genetic distances between populations. Overall, anthropogenic structures had the greatest negative impact on gene flow, whereas wetlands, grasslands, and forests imposed minimal barriers in the landscape. The most remarkable finding was the positive impact of the underpasses of the motorway A2. This element seems to be the reason why some study sites on either site of the A2 showed little genetic distance even though their habitat has been separated by a strong dispersal barrier.

**Keywords:** population genetics; landscape genetics; amphibian conservation; microsatellite analysis

## 1. Introduction

Anthropogenic influence has shaped and changed our landscape drastically. The massive expansion of agricultural land, urban areas, and the resultant soil sealing are important factors for the dramatic decline of biodiversity [1]. These features can impose dispersal barriers in a landscape that hinder the ability of individuals to migrate between habitats. Amphibians are especially affected by these changes. Since amphibians are often organized in meta-populations, habitat fragmentation has a significant impact on their ability to maintain gene flow between subpopulations and breeding sites [2]. Especially urbanized landscapes widely lack features such as steppingstone biotopes and migration corridors between important breeding sites. Therefore, habitat fragmentation is an increasing concern in species conservation.

Landscape genetics is often used as a tool in the development of conservation strategies, since the detection of gene flow barriers in a landscape and their effects on population connectivity provides information about the species interaction with their environment [3]. This method is applicable for a broad range of animal groups, including amphibians [4–6].

The global IUCN status for *Bombina variegata* is considered "least concern" [7]; however, in Germany, this species is considered "critically endangered" [8], and "threatened with extinction" in several areas [9–11]. Moreover, *B. variegata* is included in Appendix II and IV of the Flora-Fauna-Habitat-Directive, and is therefore deemed worthy of protection, including their habitats. In all three biogeographical regions of Germany (Atlantic, alpine,

and continental) the conservation status for *B. variegata* is considered "inadequate-bad" or "unfavorable-bad" [12]. The bad conservation status is the consequence of population declines due to habitat fragmentation and destruction. Their natural habitats, alluvial forests accompanied by natural floodplains, have become rare [13]. The secondary habitats of the yellow-bellied toad (quarries or military training sites) are often isolated habitats, and are cost-intensive in maintenance when production is stopped [14]. In particular, the northern populations are endangered or have gone extinct due to habitat loss [15]. Since habitats are generally more isolated in north Germany, *B. variegata* shows greater genetic isolation in the northern federal provinces of Germany than in the south [14–16].

The species is rather faithful to its breeding site, and is considered a short-distance migrant [17]. Therefore, the toads are sensitive to isolation by distance ("IBD"). The IBD model describes the distribution of genetic variation over a given geographic region, and is used in modelling of natural populations [18]. Since this method neglects spatial variations of migration and gene flow bound to landscape features, the isolation by resistance (IBR) model was designed to overcome these shortcomings. The IBT model calculates the resistance particular landscape features pose to the migration of a certain species [19]. In order to determine migration barriers in the landscape, their impact on the genetic structure and the effective distance between subpopulations, the landscape in question is categorized into its different components (wetland, grassland etc.).

The northern Weser Hills represent the northern border of distribution for *B. variegata* [15]. The associated subpopulations of the yellow-bellied toad examined in the northern Weser Hills were previously found to be genetically isolated and scattered [20]. Identifying genetic connectivity (or lack thereof) between subpopulations while simultaneously taking the role of certain landscape features into account could give an insight in the species situation in this area and help to develop measures in order to preserve one of the most vulnerable populations of the yellow-bellied toad in Germany. Due to the situation of this species in the Weser Hills, we hypothesized poor genetic connectivity in context with landscape features which present barriers to gene flow. The aims of this study were (1) determining genetic connectivity; (2) identifying the population structure; (3) evaluating the effects of geographic and effective distance on gene flow; and (4) determining landscape permeability for yellow-bellied toads in the northern Weser Hills.

## 2. Materials and Methods

### 2.1. Study Area

The 15 sites analyzed in this study are located in the northern Weser Hills in the districts of Minden-Lübbecke, Hameln-Pyrmont, and Schaumburg (Figure 1).

Most of the study sites, such as Liekwegen ("LI"), Wülpker Egge ("WE"), Segelhorst ("SH"), and Pötzen ("PA"), are man-made secondary habitats, such as quarries and military training sites (Nato-Station ("NS")), as well as sand, gravel, and clay pits (Edler/Brinkmeyer ("BM") and Fuchsloch ("FL")). Nature conservation areas can also be found in or around some study sites (many areas around Bückeberg). Some sites are declared to be nature reserves themselves (Bokshorn ("BH")). FL is also a nature reserve, and the only study site located west of the Weser. Five study sites (BM, WE, Messingsberg ("MB"), Schlingmühle ("SM") and Bernsen ("BE")) are located very close to the motorway A2 that passes through the study area (yellow line in Figure 1). Between 2013 and 2016 around 7000 larvae were released at the study sites BE, Rhoden ("RO"), SH, and WE as part of a reintroduction program of the Nature and Biodiversity Conservation Union ("NABU"), Lower Saxony [21]. These individuals stem mostly from LI, but also from BM. The reintroduced sites were excluded for the landscape genetic analysis of this study, since they could falsify the results. A total of 440 individuals were analyzed for this study. Earlier, buccal swab samples were collected from the toads during the breeding season from May to September 2016, as described in [20].

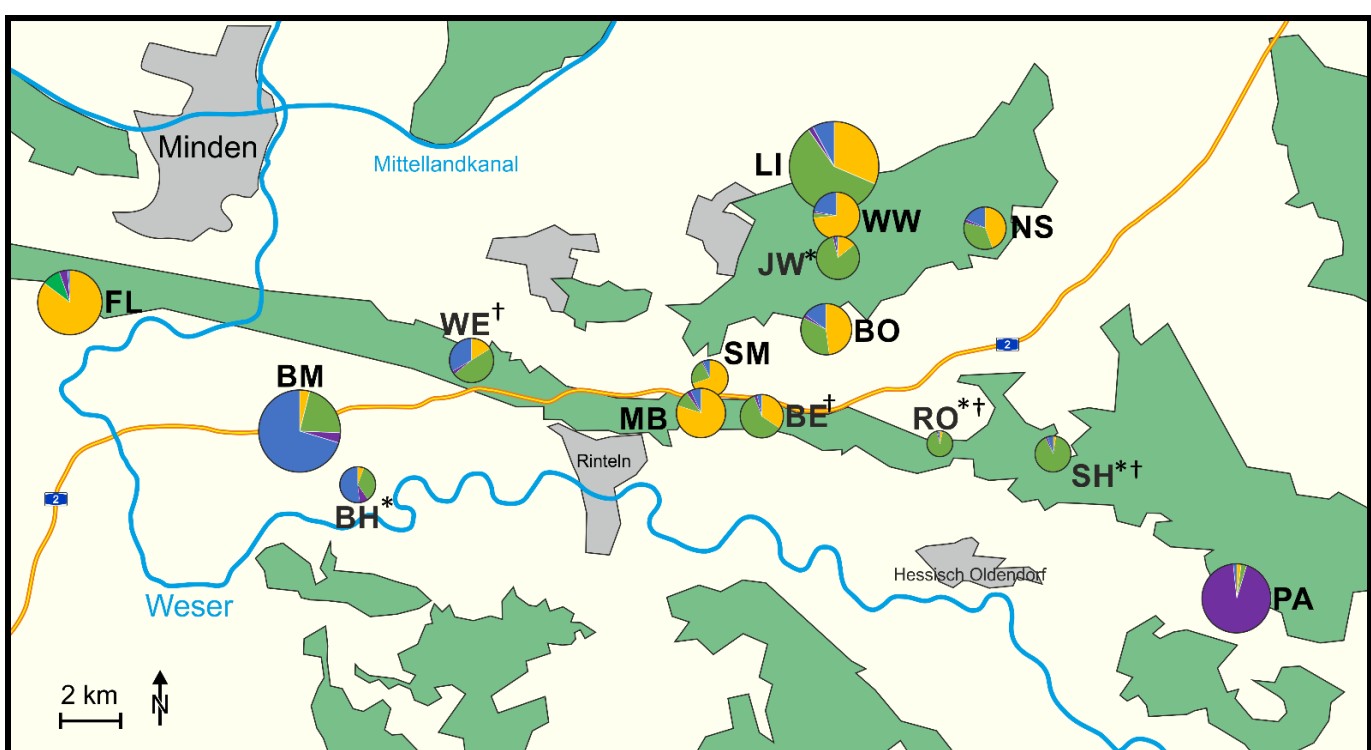

**Figure 1.** Study area, locations of the 15 sample sites in the northern Weser Hills and geographic distribution of the four estimated genetic clusters (see Results and Figure 2). Coloration indicates the membership of the individuals to the given cluster. Pie chart size is relative to the number of sampled individuals per study site. Fuchsloch ("FL"), Edler/Brinkmeyer ("BM"), Bokshorn ("BH"), Wülpker Egge ("WE"), Messingsberg ("MB"), Schlingmühle ("SM"), Bernsen ("BE"), Liekwegen ("LI"), Waldwiese ("WW"), JBF-Wiese ("JW"), Borstel ("BO"), Rohden ("RO"), Nato-Station ("NS"), Segelhorst ("SH"), and Pötzen ("PA"). * = *n* < 10. † = reintroduced populations, *n* total individuals = 440.

### 2.2. Microsatellite Genotyping

Previously published data [20,22] for nine microsatellite markers [23,24] were used for this study in order to obtain a larger dataset for the population and landscape genetic analysis. In [20], the amplification of the microsatellite B13 failed in 2016, but was successfully amplified for this study. Therefore, a total of ten microsatellite loci are collectively analyzed in this study. The polymerase chain reaction ("PCR") products of B13 were genotyped by Eurofins (fragment length analysis), and the resulting data analyzed using GeneMapper version 5 (Life Technologies). Due to differences in the genotyping methods between previous data and the current study, the fragment lengths of B13 were adjusted using the package MsatAllele [25] in the statistic program R [26].

### 2.3. Population Genetic Analysis

The genetic population structure was analyzed using the program STRUCTURE version 2.3.4 [27]. The admixture model with allele frequencies correlated and the settings 500,000 iterations after a burn-in period of 100,000 were chosen. Twenty runs for K1 to K5 were computed. The K with the highest probability was calculated according to the ΔK method [28]. To determine the geographic distances between study sites the Geographic Distance Matrix Generator version 1.2.3 [29] was used. BayesAss version 3.0.4 [30] was used to calculate migration rates, i.e., the fraction of individuals per generation that are migrants from a different population. As suggested in the manual [31], we applied 10,000,000 iterations for the Markov chain Monte Carlo with an interval of 1000 between samples and a burn-in length of 1,000,000. The acceptance rate of the migration rate was adjusted to fit within the range of 20% to 60% by changing the value of the migration rate mixing parameter from its default 0.1 to 0.55. Data visualization was carried out using the program

Circos [32]. FSTAT version 2.9.3.2 [33] was used to test for linkage disequilibrium. The R package hierfstat version 0.5–7 [34] was used to calculate pairwise and overall genetic difference ("$F_{ST}$") value [35]. Due to the difficulties in detecting null alleles [36], two different programs were used, GenePop version 4.2 [37] and ML-Nullfreq_frequency [38]. To minimize the chances for false-positive results and therefore erroneously detected null alleles, only null alleles detected with both methods were considered true null alleles.

*2.4. Landscape Genetic Analysis*

The assumed relative cost that a landcover type imposes on the movement of an individual is defined as their resistance. Elements hindering gene flow and migration, such as streets and motorways, are assigned a higher resistance value, whereas features such as grassland and soil moisture, facilitating genetic connectivity, are assigned a lower value. Based on these values, the effective distance between subpopulations can be calculated and put into context with the genetic data. A total of eight landcover types were analyzed in this study (Table 1). The layers *Imperviousness*, *Streams, Ponds & Moist Soil*, *Weser*, *Forest* and *Grassland* stem from Copernicus Land Monitoring Service 2018 (copernicus.eu). The layer *A2 Underpasses*, *Agricultural Land* and *A2 Motorway* were developed in ArcMap 10.5.1 [39], a full reference list of the layers used can be found in Supplementary Materials Table S1. The layer *Imperviousness*, originally stating values of permeability from 0 to 100% was adjusted to fit within the 0 to 10 range of resistance values by dividing the values by 10. Neglecting single layers entirely to test elements separately in regression models can erroneously implicate areas with low resistance, and falsify the effective distance calculated with the remaining layers [40]. Therefore, all layers were always kept in all conducted calculations. To test for the accuracy of the assigned values and the impact of certain elements, resistance values were individually changed to a different tier. A reference layer (REF) was developed with resistance values, according to the hypotheses of this study, to create a baseline for the testing of individual layers. Resistance values between 0 and 10 were assigned to these features in ArcMap using the toolbox Spatial Analyst. All elements were categorized in four tiers: habitat (resistance of 0), favorable matrix (3), less favorable matrix (6), and strong barrier (9). The values for the reference layer were chosen with consideration of previous literature about the impact of landscape elements on amphibians and *B. variegata*. Soil moisture can increase amphibian migration [41], therefore moist soil is classified as a habitat (0). Grassland and forest area was both considered a favorable matrix and not a habitat, since these features can vary in their permeability due to forest type and predation pressure. Agricultural land, especially monoculture plantations can negatively impact migration [14,15]. Due to its depth and velocity, the river "Weser" in this area was considered a strong dispersal barrier [6], regardless of differing width of the water body [42, personal communication]. The same was applied for the motorway A2. Since there is no reference literature on the resistance value of underpasses, these were considered as a less favorable matrix.

**Table 1.** Landscape categories with resistance tier list and corresponding resistance values for REF layer. References: [6,14,15,41,42]. All raster layer (except *Agricultural Land*, *A2 Motorway* and *A2 Underpasses*) © by European Union, Copernicus Land Monitoring Service 2018, European Environment Agency, copernicus.eu [Accessed: 1 June 2020].

| Landscape Category | Resistance Tier | Assigned Resistance Value |
|---|---|---|
| *Streams, Ponds & Moist Soil* [6,41] | Habitat | 0 (reduces resistance value of *Forest* and *Grassland* to 0 when overlapping) |
| *Forest* (dry) [6,15] | Favorable matrix | 3 |
| *Grassland* (dry) [6,15] | Favourable matrix | 3 |
| *Agricultural Land* [14,15] | Less favorable matrix | 6 |

**Table 1.** *Cont.*

| Landscape Category | Resistance Tier | Assigned Resistance Value |
|---|---|---|
| *A2 Underpasses* | Less favorable matrix | 6 |
| *Weser* [6,42] | Strong barrier | 9 |
| *A2 Motorway* [6,14] | Strong barrier | 9 |
| *Imperviousness* [14] | Habitat to strong barrier | 0% to 100%, adjusted to fit within 0 to 10 range |

Next, nine models/resistance hypotheses were developed, each containing one landscape layer with a resistance value deviating from the reference layer (Table 2).

**Table 2.** Landscape resistance models.

| Model ID | Model | Explanation |
|---|---|---|
| 1 | UNDIF | Resistance calculated using an undifferentiated landscape |
| 2 | REF | Reference resistance values listed in Table 1 |
| 3 | UNDER3 | *Underpasses A2* with resistance value 3 |
| 4 | UNDER9 | *Underpasses A2* with resistance value 9 |
| 5 | AGRI3 | *Agricultural Land* with resistance value 3 |
| 6 | AGRI9 | *Agricultural Land* with resistance value 9 |
| 7 | WET0 | *Streams, Ponds & Moist Soil* do not reduce resistance value of layers *Grassland* and *Forest* from 3 to 0 |
| 8 | WESER3 | *Weser* with resistance value 3 |
| 9 | WESER6 | *Weser* with resistance value 6 |

The pairwise $F_{ST}$ values and migration rates calculated with BayesAss were chosen as dependent variables. These genetic distances stem from a total of 440 individuals, see Table 3 for information on habitat types and number of individuals tested at each study site. After assigning resistance values to the different elements, all layers were combined to form a resistance map. The pairwise effective resistance between subpopulations (focal nodes) was calculated using Circuitscape version 5 [43]. The same program was used to produce current density maps. In this depiction, the potential movement of individuals through the landscape is interpreted as an electrical current traveling between the focal nodes through each raster cell. The lower the resistance from the landscape, the higher the current density in this area. A high current density is shown with yellow colors, indicating that the probability of successful movement and gene flow are highest in these areas.

To compare the various landscape genetic models, we used maximum-likelihood population-effects models ("MLPE") with the R package ResistanceGA [44]. This approach has been shown to perform well in discriminating competing landscape resistance models [45]. The R package Adegenet Version 1.3-1 [46] was used for the conversion of the genetic data. MLPE, with the model fit indices Bayesian Information Criterion (BIC) and marginal $R^2$, calculated using PiecewiseSEM [47], were chosen in this study since the method outperformed other regression methods and indices [45,48]. When applied to simulated populations BIC showed the most accurate results, while the marginal $R^2$ had drawbacks due to the bias towards more complex models of the simulation [45]. However, in combination with other indices, this method is suitable for the detection of the best fit model. The likelihood ratio test for the nearest model was carried out with the R package lmtest [49] (function lrtest with log-likelihood, degrees of freedom, likelihood ratio Chi-squared statistic, and corresponding *p* value). The resistance between the given focal nodes is expected to increase with increasing distance regardless of the landscape elements and their resistance value. Therefore, an UNDIF variable was included in the

calculations, representing the isolation-by-distance. Additionally, a partial Mantel test to test for the influence of isolation by distance on the genetic distance and migration rates was performed in R using the package vegan version 2.5–7 [50].

**Table 3.** Summary of diversity indices for yellow-bellied toads in northern Weser Hills at 15 sample sites. Sample size ("*n*"), gene diversity ("Gd"), expected heterozygosity ("$H_e$"), observed heterozygosity ("$H_o$"), fixation/inbreeding coefficient ("$F_{IS}$"), number of private alleles ("Np"), allelic richness ("Ar"), standard deviation ("SD"), * = $n < 10$. † = reintroduced populations, *n* total individuals = 440.

| Sample Site | Habitat | Distance to Next Population (km) | N | Gd | $H_e$ | $H_o$ | $F_{IS}$ | Np | Ar |
|---|---|---|---|---|---|---|---|---|---|
| FL | Former clay pit, nature reserve | 8.68 | 12 | 0.42 | 0.42 | 0.38 | 0.05 | 1 | 1.86 |
| BM | Sand/gravel pit | 2.33 | 85 | 0.48 | 0.48 | 0.35 | 0.26 | 5 | 2.08 |
| BH * | Former sand pit, nature reserve | 2.33 | 4 | 0.4 | 0.4 | 0.43 | −0.20 | 0 | 1.81 |
| WE † | Active quarry | 5.19 | 14 | 0.49 | 0.49 | 0.37 | 0.21 | 2 | 2.09 |
| MB | Active quarry | 0.68 | 47 | 0.51 | 0.50 | 0.43 | 0.15 | 2 | 2.11 |
| SM | Farm track/wheel tracks/stepping stone | 0.68 | 10 | 0.49 | 0.48 | 0.34 | 0.26 | 1 | 2.06 |
| BE † | Inactive quarry | 2.56 | 31 | 0.47 | 0.47 | 0.39 | 0.16 | 3 | 2.05 |
| LI | Nature reserve, inactive quarry | 1.06 | 106 | 0.54 | 0.54 | 0.42 | 0.21 | 5 | 2.23 |
| WW | Forest meadow | 1.06 | 12 | 0.59 | 0.65 | 0.49 | 0.12 | 2 | 2.3 |
| JW * | Forest meadow | 1.41 | 7 | 0.51 | 0.50 | 0.41 | 0.12 | 0 | 2.03 |
| BO | Former clay pit, nature reserve | 1.81 | 17 | 0.47 | 0.47 | 0.44 | 0.04 | 2 | 2.05 |
| RO *† | Inactive quarry | 2.94 | 2 | 0.58 | 0.52 | 0.4 | −0.03 | 0 | 2.1 |
| NS | Military training site | 5.69 | 24 | 0.53 | 0.53 | 0.48 | 0.07 | 3 | 2.21 |
| SH *† | Active quarry | 2.94 | 6 | 0.41 | 0.40 | 0.37 | 0.004 | 0 | 1.89 |
| PA | Inactive quarry | 7.65 | 63 | 0.39 | 0.39 | 0.33 | 0.15 | 10 | 1.85 |
| Mean/Total (± SD) | - | - | 440 | 0.49 | 0.48 | 0.4 | 0.1 | 2.4 | 2.05 |
| | | | - | (±0.06) | (±0.06) | (±0.05) | (±0.12) | (±2.58) | (±0.14) |

## 3. Results

### 3.1. Population Genetic Diversity

A total of seven of the ten analyzed microsatellites were polymorphic for all 15 study sites (Supplementary Materials Tables S2 and S3). 1A was monomorphic at seven study sites (FL, BH, Waldwiese ("WW"), Borstel ("BO"), RO, NS and SH), and F22 at four sites (BH, SM, JBF-Wiese ("JW") and SH). B13 turned out to be monomorphic only at RO; however, only two individuals were examined at this site. A total of 123 alleles were found, including 36 private alleles. On average, 12.3 alleles and 2.4 private alleles were analyzed per study site. No loci at any study site showed evidence for a linkage disequilibrium or null alleles. A total of 22 significant deviations from the HWE (*p*-value < 0.05) were detected, with almost half of these found at F2, and six at B13. The gene diversity ranged from 0.59 (WW) to 0.39 (PA) with an average of 0.49 (Table 3).

The expected heterozygosity was generally higher than the observed for all study sites, BH being the only exception. Only four out of the 15 study sites showed no private alleles. The standard deviation of the fixation/inbreeding coefficient ("$F_{IS}$ value") is ±0.12, resulting in fluctuating values. This is probably due to the difference in sample size. Sites

with low numbers of analyzed individuals (BH, RO, SH) generally showed a lower $F_{IS}$ value. In nine cases, the inbreeding coefficient was higher than the average of 0.1, with BM, SM and LI showing the overall highest $F_{IS}$ values. Additionally, both LI and BM showed five private alleles. Moreover, PA showed the highest number of private alleles and a low allelic richness.

### 3.2. Population Structure and Migration

The STRUCTURE analysis revealed that the most likely number of genetic clusters for yellow-bellied toads in the northern Weser Hills was K = 4 (Figure 2a,c).

G4 was very clearly demarcated against all other clusters, with PA being the only study site within this cluster. LI and other study sites around Bückeberg showed no distinct separation, but rather a mixture between cluster G1 and G2. The pairwise $F_{ST}$ ranged from 0.01 to 0.33 (Supplementary Materials Table S4), with a global $F_{ST}$ of 0.1. PA showed overall great genetic distance (categorized according to [18]) to all study sites, and very great to FL and SH. BM and BH showed moderate to very great genetic differences to almost all study sites, except for WE. The sites WE, BE, RO, and SH were reintroduced, and showed overall comparable smaller genetic difference to each other and the populations around Bückeberg, especially LI. Although the structure analysis showed that FL is assigned to cluster G1, the geographical distance between study sites in this cluster was rather large (20 km between FL and MB). The Mantel test indicated a moderate correlation between the genetic and the geographic distance (r = 0.52, *p* = 0.01). The Circos plot in Figure 3 shows the migration rates per last generation between each location (Supplementary Materials Table S5).

Estimated migration rates between study sites using BayesAss3 showed that most migration towards other study sites was found around LI. With the exception of MB, there was almost no migration towards this site; however, individuals appear to have migrated from LI into other study sites, mostly WE, BE, and SH. PA displayed the lowest levels of migration, which further indicates the genetic isolation of this study site. The data suggests that a large fraction of individuals emigrated from the NS population to the study sites WW, JW, and BO. All of these sites are in close proximity to each other. Moreover, high migration rates were found from BO to FL and SM, despite the long geographic distance between these study sites.

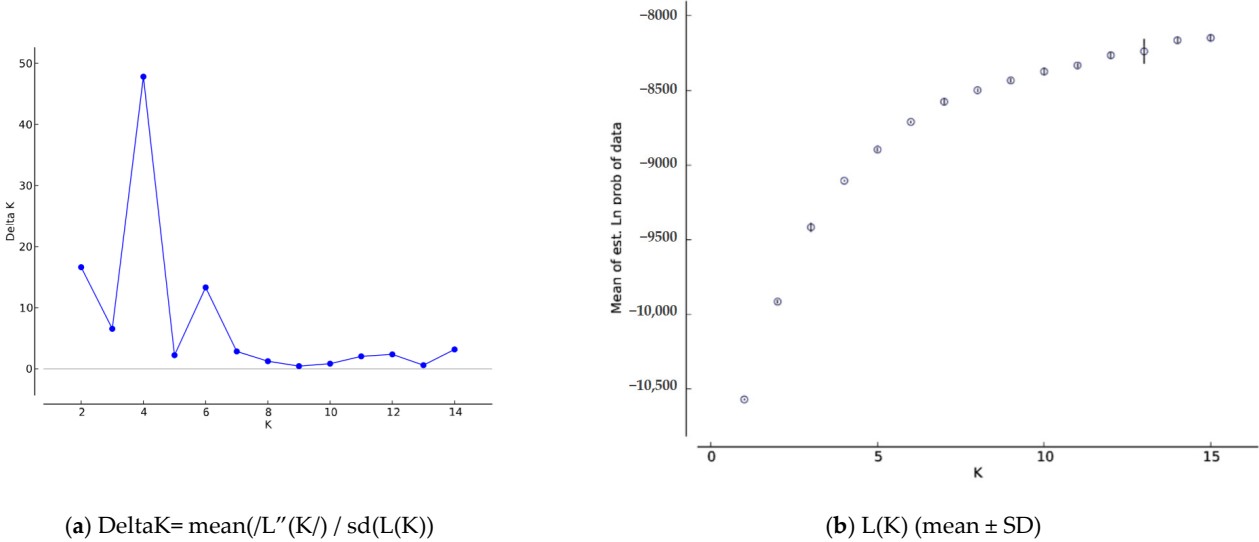

(**a**) DeltaK= mean(/L″(K/) / sd(L(K))　　　　　　　　　　　(**b**) L(K) (mean ± SD)

**Figure 2.** *Cont.*

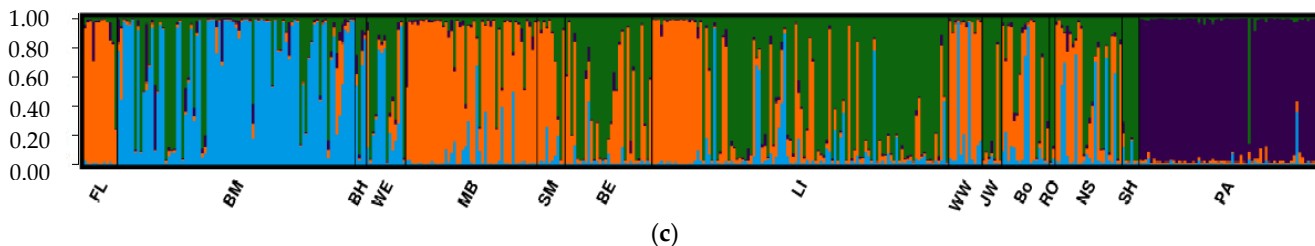

(**c**)

**Figure 2.** Detecting the most likely number of genetically distinct groups within yellow-bellied toads in the northern Weser Hills based on (**a**) mean L(K) ± SD over 20 runs per K, (**b**) Maximum likelihood estimation of K [30] and (**c**) percentage of population assignment of K = 4. Each individual is visualized as a single line. Coloration indicates the membership of the individual to the given cluster ("G"). G1 (orange) = FL, WW, NS, BO, SM, MB. G2 (green) = WE, LI, JW, BE, RO, SH. G3 (blue) = BM, BH. G4 (purple) = PA.

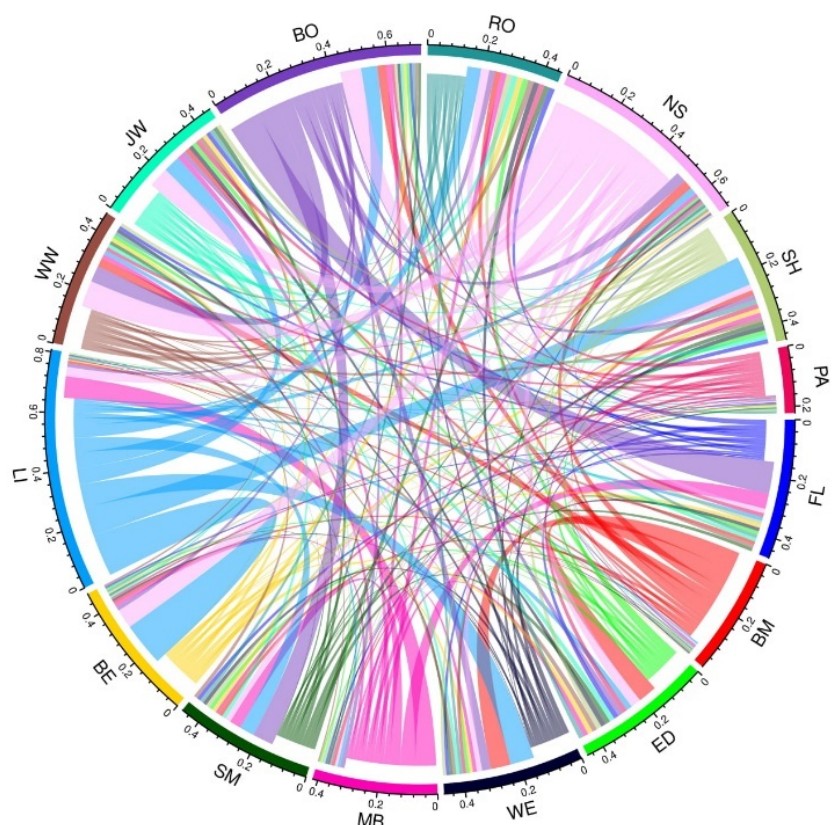

**Figure 3.** Circos plot of inferred contemporary migration rates (per last generation) between subpopulations of yellow-bellied toads based on 10 microsatellites. Plot corresponds to the detected migration (full values from BayesAss3 are provided in Supplementary Materials Table S5). Width of the curves indicates the amount of migration, according to the scale, going from one subpopulation into the other.

*3.3. Landscape Genetic Analysis*

The reintroduced study sites BE, RO, SH, and WE were excluded from all further calculations, since the gene flow between these sites is artificial. The landscape models revealed the undifferentiated landscape (UNIDF), as the best fit model according to the BIC value (Table 4, based on effective distances in Supplementary Materials Table S6).

**Table 4.** Model selection analysis with delta Bayesian Information Criterion ("BIC") and marginal $R^2$ as indicators showing the effects of different landscape structure models on population differentiation ($F_{ST}$) and migration rate ("mig") for pairs of *B. variegata* subpopulations. LogLik for complex models was calculated between each model and an undifferentiated landscape ("UNDIF"). * indicates significant Pr values < 0.05.

| Model | Delta BIC | Marginal $R^2$ | LogLik for Complex Model | Chisq | Pr (>Chisq) |
|---|---|---|---|---|---|
| | | | $F_{ST}$ values | | |
| $F_{ST}$~UNDIF | 0 | 0.001 | 89.389 | | |
| $F_{ST}$~REF | −1.25 | 0.292 | 92.772 | 6.767 | 0.034 * |
| $F_{ST}$~UNDER3 | −1.07 | 0.299 | 92.862 | 6.947 | 0.031 * |
| $F_{ST}$~UNDER9 | −1.25 | 0.292 | 92.772 | 6.767 | 0.034 * |
| $F_{ST}$~AGRI3 | −4.18 | 0.032 | 91.305 | 3.832 | 0.147 |
| $F_{ST}$~AGRI9 | −2.24 | 0.324 | 92.276 | 5.774 | 0.056 |
| $F_{ST}$~WET0 | −1.17 | 0.294 | 92.811 | 6.844 | 0.033 * |
| $F_{ST}$~WESER3 | −1.49 | 0.283 | 92.652 | 6.526 | 0.038 * |
| $F_{ST}$~WESER6 | −1.34 | 0.288 | 92.726 | 6.675 | 0.036 * |
| | | | Migration rates | | |
| MIG~UNDIF | 0 | 0.08 | 122.08 | | |
| MIG~REF | −0.99 | 0.179 | 125.593 | 7.028 | 0.023 * |
| MIG~UNDER3 | −0.41 | 0.19 | 125.879 | 7.6 | 0.022 * |
| MIG~UNDER9 | −0.99 | 0.179 | 125.593 | 7.028 | 0.03 * |
| MIG~AGRI3 | −4.60 | 0.085 | 123.788 | 3.418 | 0.181 |
| MIG~AGRI9 | −0.42 | 0.19 | 125.876 | 7.593 | 0.022 * |
| MIG~WET0 | −1.02 | 0.178 | 125.576 | 6.994 | 0.03 * |
| MIG~WESER3 | −0.93 | 0.18 | 125.62 | 7.08 | 0.03 * |
| MIG~WESER6 | −0.97 | 0.179 | 125.604 | 7.05 | 0.03 * |

Therefore, isolation-by-distance has a correspondingly high impact on both the genetic distance as well as the migration rate. This is in line with the result of the Mantel test. However, the pairwise genetic distances and migration rates cannot be explained solely by the geographic distances between study sites, since the marginal $R^2$ was overall lowest when the landscape-free undifferentiated model was used. The $R^2$ value shows that most of the genetic structure, using both dependent variables, can be explained with the models AGRI9 and UNDER3, suggesting that agricultural land imposes a strong barrier for the toads, and the underpasses of the A2 are considered a favorable matrix. The Mantel test analyzing the effective distance (supplementary Materials Table S6) of the reference values showed a highly significant positive correlation with the migration rates calculated with BayesAss3 (r = 0.9, $p = 9.999^{-5}$). However, the partial Mantel test showed a significant correlation between migration rate and effective distance after the influence of geographic distance was eliminated (r = 0.44, $p = 0.01$). The genetic distance showed no such significant correlation when substituted for the migration rate (r = 0.356, $p = 0.114$).

All MLPE models showed higher log-likelihood than the UNDIF model, UNDER3 having the highest likelihood and significance. The least fitting model was AGRI3, where the agricultural area was considered a favorable matrix. This was indicated through both model selection indices (high delta BIC value and low $R^2$), as well as the low model likelihood and significance. Clearly, the models assigning high resistance values to anthropogenic structures performed well (such as ARGI9 or REF), whereas models such as ARGI3 performed worse, both in context with the $F_{ST}$ values as well as the migration

rates. The models WET0, WESER3, and WESER6 all showed a significant higher likelihood over the undifferentiated landscape and performed overall well. However, only the study site Fuchsloch was affected by the dispersal barrier that the Weser imposes in this area. The REF model was used to calculate a current density map of the northern Weser Hills (Figure 4).

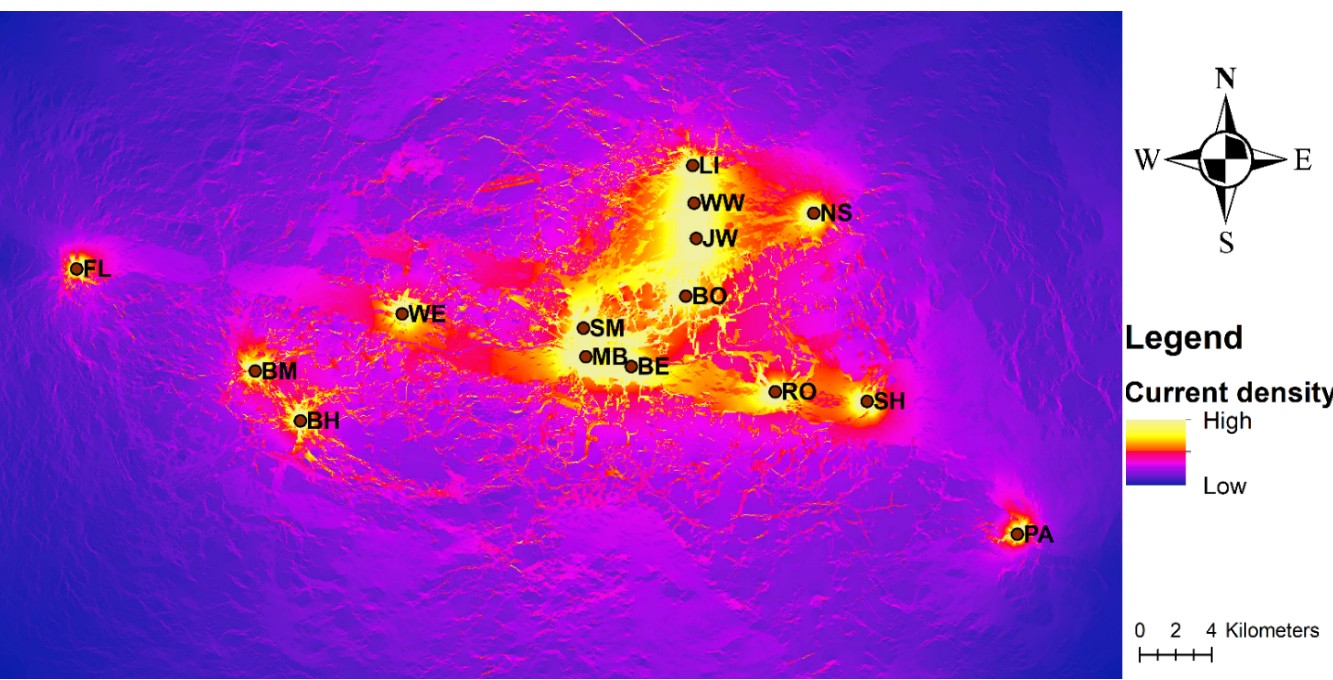

**Figure 4.** Current density map of the northern Weser Hills created with Circuitscape and ArcMap. Current density between 15 focal nodes is shown.

The highest current density (Yellow, Figure 4) was found between study sites in close proximity to each other (such as the cluster around Bückeberg, G2) and between populations not separated by agricultural land or urban areas. The "Kamm des Wesergebirges", located between the areas WE and MB, seems to be a corridor for gene flow in this area. The current density here is moderate between study sites. "Kamm des Wesergebirges" mainly consists of broadleaf forests surrounded by agricultural area. However, this corridor is interrupted by the Weser and parts of the city Minden. Therefore, the study site Fuchsloch is largely isolated from the other sites. The current density between the focal node PA and the next study site in this area is low and manly consists of agricultural land and forests.

## 4. Discussion

In this study we combined genetic data of *B. variegata* with landscape elements in the northern Weser Hills. Overall, the results suggest that most examined populations in this area are in poor genetic condition and show moderate to great genetic differentiation. The detected genetic differentiation was due to both isolation-by-resistance and isolation-by-distance, since the UNDIF model performed best and the Mantel test showed corresponding correlations, but the $R^2$ was overall higher in all developed resistance models. Anthropogenic structures, especially agricultural land, had a negative impact on genetic connectivity.

### 4.1. Genetic Diversity and Population Structure

With the addition of the microsatellite B13 to the data set and increasing the number of individuals by combining two datasets [20,22], a total of four genetic clusters were found compared to the previous study, with three clusters examining the same study sites [20].

This shows how broader range of loci in the sample pool and increasing sample size can improve the analysis of genetic structures. In this case, a further distinction between cluster G1 and G2 was detected. The poor genetic connectivity found in the northern Weser Hills is in line with previous studies [51]. The study sites are located on the northern border of distribution of *B. variegata*, with only few subpopulations. A study including genetic data derived from numerous *B. variegata* occurrences in Germany showed that the northern Weser Hills form a separate cluster to the other localities in Germany [22]. Within the context of *B. variegata* occurrences in Germany, all study sites in the northern Weser Hills, with the exception of PA, belong to the northern cluster, the others form the wide, southern cluster. However, some study sites showed good genetic connectivity (e.g., sites around Bückeberg), probably due to the fact that yellow-bellied toads are short-distance migrators. Although there are recorded exceptions to this statement [52], the activity radius of this species declined drastically after 300 m [17]. Therefore, study sites in close proximity to each other are more likely to maintain better connectivity, than far apart study sites. Genetic connectivity between subpopulations is essential for their fitness and survival, especially in amphibians [2]. A significant positive correlation between genetic and geographic distance was found in this study. A distance between 1 and 2 km is recommended for a good connectivity between subpopulations of the yellow-bellied toad [53], as well as the implementation of steppingstone biotopes in between subpopulations 5 or more km apart [16]. All these findings and recommendations emphasize the importance of short scale reestablishment of connectivity in the conservation of this species. The conservation goal for *B. variegata* is to establish metapopulations with >1000 connected individuals [54].

The study site PA showed a high genetic distance to all other study sites, and was furthermore determined as a separate cluster. PA is separated from the other sites by agricultural land and an approximately 7 km long strip of coniferous forest. This forest type in combination with the long distance between study sites could be one reason for the lack of gene flow. With the inclusion of genetic data derived from study sites further south of the northern Weser Hills, ref. [22] PA was assigned to the second, southern German cluster.

Some discrepancies were observed at the study site FL, for example the detected, but unrealistic long-distance migrations. The toads of this study site showed high migration rates to other study sites, regardless of the geographic distance or landscape barriers, which seems unrealistic. The STRUCTURE analysis showed an affiliation of FL to the G1 cluster. The closest study site to FL within the G1 cluster is MB, which is approximately 29 km away. The nearest study site is approximately 8.7 km away (BM, G3). Both impose unlikely migration pattern for *B. variegata*. However, it is possible that small groups of this species do exist between these study sites, making a natural gene flow possible. It is also possible that unauthorized translocations of toads have altered the gene flow artificially in this area. The comparably long life cycle of *B. variegata*, with reproductive activity up until old age, leads to overlapping generations, meaning, that more than one breeding generation is present in a mating system [15]. Therefore, an individual migrating (or being translocated) into a previously isolated population and reproducing successfully can have a larger relative contribution on the gene flow than the estimated migration rate would indicate [55]. For further evaluation of the true migration pattern and demographic connectivity, the capture-recapture method is advised.

The results from this study indicate that the study site WE shows similarities to the BM/BH cluster. Considering that the cluster BM/BH and the rest of the northern Weser Hills are separated by roughly 5 km of agricultural land, imposing a barrier for yellow-bellied toads, this gene flow is most likely artificially caused by the reintroduction of toads from LI and partly BM to other study sites in 2013 to 2016, including WE.

*4.2. Effects of Geographic and Resistance Distances on Gene Flow*

The effects of both geographical and effective distance on the genetic connectivity of the analyzed subpopulations of *B. variegata* are apparent. The models with the best

fit indicated negative effects of anthropogenic structures (streets, agricultural land, and urban areas) on the genetic connectivity of the toads in the area, such as AGRI9 and REF. This is as expected, since these elements are already established as gene flow inhibitors for *B. variegata* [6,14,15,53]. However, the strong impact of underpasses on gene flow was noticeable. The genetic distance as well as the migration rates calculated for the study sites on either side of the motorway were not largely affected by the motorway, most likely due to the underpasses having a positive impact on the connectivity in yellow-bellied toad populations, and improving the otherwise missing permeability of this motorway. These findings could potentially be relevant for the planning and development of green infrastructure. This term describes the implementation of strategies that benefit the persistence of biodiversity into planning policies [56]. Further testing with capture-recapture experiments and GPS-tracking should be applied at study sites located either side of the A2, connected by an underpass (WE to BM/BH or the Bückeberg cluster to MB and BE) to verify these findings.

A landscape genetic study targeting the endangered goitered gazelles using the method of Mantel test found a decrease of relatedness between populations with increasing geographic distance [57]. During our study, we found similar results, with the method of Mantel test showing better model fit of IBD over IBR in contrast to the use of MLPE. A study conducted for the blotched tiger salamander showed a strong IBD influence, but a better model fit when landscape elements were included [5]. Implementing resistance models into the analysis can broadly improve the understanding of genetic connectivity in a landscape, and how certain elements can influence connectivity which was also the case for our study. Using the MLPE method a study on two salamander species, *Rhyacotriton kezeri* and *Rhyacotriton variegatus,* showed that forest coverage is essential for the dispersal of these species, while fragmentation and deforestation have an overall negative impact on the genetic connectivity [58]. Interestingly, in this study the calculations were separated via clusters and IBD was selected as the most fit model for the northern population clusters. It is possibly advisable for larger populations to implement different models for clusters, especially when the habitats of these clusters differ drastically, or the study area is very large.

Implementing the analysis of movement pattern with GPS tracking could improve the accuracy of the assignment of resistance value, and would give more insight of the true movement pattern of the species. Moreover, studies within the field of landscape genetics typically vary regarding their applied methods [45]. Further investigation within this area will hopefully standardize and unify the applied methods to improve comparability between studies. However, landscape genetic analysis still is a powerful tool in evaluating strategies in species conservation.

In conclusion, except for one, all hypotheses concerning the effect of the landscape elements on the resistance values for *B. variegata* migration stated at the beginning of this study were accepted. The results of this study could not distinctly prove that soil moisture improves gene flow in yellow-bellied toads.

## 5. Conclusions

Considering the distances travelled of the toads and the comparably dense landscape structure of the northern Weser Hills, the best method to improve the situation is to decrease the geographic distance between populations. Adding steppingstone habitats to the landscape, installing more motorway underpasses, and creating new habitats would therefore benefit genetically isolated populations. Moreover, agricultural land was proven to impose a highly negative impact on this species. Intensive agriculture is a major threat to biodiversity, and adds to the impermeability of the landscape, resulting in decreased migration potential [59]. Wider grass verges between fields and steppingstone biotopes between isolated populations more than 5 km apart [16] can help loosen up the landscape and make it more permeable for amphibians. An additional aspect is the embarkment of rivers. In many places, *B. variegata* is already forced into secondary habitats, such as

quarries and military training fields [15]. Both the desired landscape permeability and the creation of suitable habitats would be achieved by allowing rivers to overflow periodically, resulting in natural floodplains.

**Supplementary Materials:** The following are available online at https://www.mdpi.com/article/10.3390/d13120623/s1. Table S1: Reference list of used layers from Copernicus Land Monitoring Service © and maps created in ArcMap to form the landscape models for the analysis of *B. variegata* in the northern Weser Hills. Table S2: Summary statistics for the used microsatellite markers (loci), averaged across all study sites. Number of detected alleles, expected heterozygosity ("$H_e$"), observed heterozygosity ("$H_o$"), size range of each locus in base pairs, allelic richness of each loci and confidence interval ("CI"). "Tetra" and "di" refers to the number of repeats (tetranucleotide or dinucleotide). 10 loci in 440 *Bombina variegata* individuals analyzed. Table S3: Genotyping results from *Bombina variegata* from each sample site in the northern Weser Hills. Table S4: Matrix of pairwise genetic ($F_{ST}$ values; lower triangle) and geographic distance (km; upper triangle) among the northern Weser Hills populations. Divided by clusters. Fuchsloch ("FL"), Edler/Brinkmeyer ("BM"), Bokshorn ("BH"), Wülpker Egge ("WE"), Messingsberg ("MB"), Schlingmühle ("SM"), Bernsen ("BE"), Liekwegen ("LI"), Waldwiese ("WW"), JBF-Wiese ("JW"), Borstel ("BO"), Rohden ("RO"), Nato-Station ("NS"), Segelhorst ("SH"), and Pötzen ("PA"). * = $n < 10$. † = reintroduced populations. Table: S5: Mean migration rate ("m[i,j]"), the fraction of individuals in population "i" that are migrants derived from population "j" (per generation) conducted with BayesAss3. Self-migration is marked grey, the 5% highest migration rates are in green, the lowest 5% in pink (self-migration was excluded). * = $n < 10$. † = reintroduced populations. Table S6: Effective distance calculated using Circuitscape for *Bombina variegata* in the northern Weser Hills.

**Author Contributions:** Conceptualization, J.K., N.B., H.P.; methodology, validation, supervision, N.B., H.P.; resources, project administration, funding acquisition, H.P.; formal analysis, data curation, investigation, writing—original draft preparation J.K.; writing—review and editing, visualization, J.K, N.B., H.P. All authors have read and agreed to the published version of the manuscript.

**Funding:** This research received no external funding outside the Institute of Zoology of the University of Veterinary Medicine, Hannover, Germany.

**Institutional Review Board Statement:** Not applicable.

**Informed Consent Statement:** Not applicable.

**Data Availability Statement:** The data presented in this study are openly available as Supplementary Materials.

**Acknowledgments:** The authors would like to thank the Nature and Biodiversity Conservation Union Germany, especially Mirjam Nadjafzadeh and Falk Eckhardt. We would also like to thank Ariel Rodríguez, Sönke von dem Berg, Katharina Westekemper, and Sebastian Wöhle. The publication was supported by Deutsche Forschungsgemeinschaft and University of Veterinary Medicine Hannover, Foundation within the funding program Open Access Publishing.

**Conflicts of Interest:** The authors declare no conflict of interest.

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
