# Peer review of "Landscape Genetics of the Yellow-Bellied Toad (Bombina variegata) in the Northern Weser Hills of Germany"

_diversity, doi:10.3390/d13120623_

Round 1

Reviewer 1 Report

This is a nice study on the population genetics of the yellow-bellied toad (Bombina variegata) in N Germany, where populations are fragmented and endangered. The study is well designed and executed and provides relevant information for the management of the species in Germany. Below I list some minor issues that the authors should consider in preparing a revised version, I hope they are useful:

1) Introduction, lines 46-50. The last two sentences are a bit awkward, it is not clear why the gazelle study needs to be highlighted here. There are many examples of landscape genetics studies in many organisms, so I think it's more appropriate to cite a general review in which studies from different animal groups are summarized. You can link this with the next sentence, for instance: “Landscape genetics has been applied in a broad range of animal groups, including amphibians” or similar.

2) Introduction, lines 51-52: I think it's more appropriate to start the paragraph mentioning the global IUCN status for the species: Least Concern, then highlight that populations are declining, especially in western and northern parts of its range, including Germany.

3) Introduction, line 65: "influence" used twice in the same sentence, please reword.

4) Introduction, line 71: what does "dispersed" mean? you mean "disperse"?

5) Materials and Methods, Fig. 1: it would be nice to add the Weser river and motorway A2 in the map. Also consider adding pie charts with colors representing average genetic ancestry proportions for each sampled locality, to help visualize genetic structure

6) Materials and methods, lines 96-99: I think you should explain in more detail how many of the samples analyzed in this new study had already been genotyped in previous studies. Does the new data comprise only genotyping at locus B13 samples that had all been previously genotyped at the other 9 loci?

7) Materials and methods, lines 105-106: did you use the model with allele frequencies correlated? also, do you mean you used the LOCPRIOR option?

8) Material and methods, lines 170-171: this sentence is awkward, something seems to be missing.

9) Results, lines 229-230: Figure 2 is labelled as Figure 1.

10) Results, lines 233-234: I think it's more appropriate to qualify genetic distances as "high" rather than "great". Please check throughout the text.

11) Results, lines 247-249: what do you mean "or lack thereof"? how can one population simultaneously concentrate most migration and "lack thereof"? Also, is the rate of self-migration the same as the proportion of individuals classified as residents by BayesAss? At any rate, I would avoid the use of "self-migration", which can be misleading.

12) Results, line 263 (Legend to table 4): some information seems to be missing; the last sentence seems incomplete.

13) Results, line 279: delete second comma in "Clearly, the models, assigning" (replace by "Clearly, the models assigning...".

14) Discussion, line 300: see comment above on the use of "great".

15) Discussion, line 305: "increasing", not "incresaing".

16) Discussion, line 306: see comment above on clarifying the number of individuals genotyped in previous studies: you used the same individuals or more?

17) Supp. Material, line 396: What do allelic richness estimates refer to, are they population averages? Please explain

Author Response

Dear Reviewer,

Herewith we submit a revised version of our manuscript entitled “Landscape genetics of the Yellow-Bellied Toad (Bombina variegata) in the Northern Weser Hills of Germany”. We thank the referees, reviewers and the editors for their comments that helped to improve the text. We did our best to correct all mistakes listed by the reviewers.

The revised version was not submitted elsewhere and we hope it is acceptable for publication in Diversity in its current form. We adhere to the guidelines for the use of animals in research and the laws of the country where the research was conducted. The manuscript contains only material that is either original or stems from publications identified by a reference. All authors have agreed to be listed, approved the final manuscript and take responsibility for its contents.

Sincerely,

Jasmin Kleißen

Point 1: Introduction, lines 46-50. The last two sentences are a bit awkward, it is not clear why the gazelle study needs to be highlighted here. There are many examples of landscape genetics studies in many organisms, so I think it's more appropriate to cite a general review in which studies from different animal groups are summarized. You can link this with the next sentence, for instance: “Landscape genetics has been applied in a broad range of animal groups, including amphibians” or similar.

Response 1: Changes have been made accordingly and a review paper targeting amphibians was included, line 47:

“This method is applicable for a broad range of animal groups, including amphibians [4-6].”

Point 2: Introduction, lines 51-52: I think it's more appropriate to start the paragraph mentioning the global IUCN status for the species: Least Concern, then highlight that populations are declining, especially in western and northern parts of its range, including Germany.

Response 2: Changes have been made accordingly, line 52:

“The global IUCN status for Bombina variegata is considered “least concern” [7], however in Germany this species is considered “critically endangered” [8], and “threatened with extinction“ in several areas [9-11].”

Point 3: Introduction, line 65: "influence" used twice in the same sentence, please reword.

Response 3: Changes have been made accordingly, line 65:

“Since this method leaves out spatial variations of migration and gene flow bound to landscape features, the isolation by resistance (“IBR”) model was designed to overcome these shortcomings. The IBT model calculates the resistance particular landscape features pose to the migration of a certain species [19].

Point 4: Introduction, line 71: what does "dispersed" mean? you mean "disperse"?

Response 4: The word “dispersed” was changed to “scattered” (line 74).

Point 5: Materials and Methods, Fig. 1: it would be nice to add the Weser river and motorway A2 in the map. Also consider adding pie charts with colors representing average genetic ancestry proportions for each sampled locality, to help visualize genetic structure

Response 5: Please see the improved figure in the manuscript.

Point 6: Materials and methods, lines 96-99: I think you should explain in more detail how many of the samples analyzed in this new study had already been genotyped in previous studies. Does the new data comprise only genotyping at locus B13 samples that had all been previously genotyped at the other 9 loci?

Response 6: The microsatellites 1A, F2, 5F, 8A, 9H, 10F, 12F, B14 and F22 have been genotyped for the studies Oswald et al. (2020) and Pröhl et al. (2021). The datasets were combined and the failed microsatellite B13 was added for this study. The respective section was reworded in the manuscript (line 110):

“Previously published data [20;22] for nine microsatellite markers [23,24] were used for this study in order to obtain a larger dataset for the population and landscape genetic analysis. In [20] the amplification of the microsatellite B13 failed in 2016 but was successfully amplified for this study. Therefore, a total of ten microsatellite loci are collectively analysed in this study.”

Point 7: Materials and methods, lines 105-106: did you use the model with allele frequencies correlated? also, do you mean you used the LOCPRIOR option?

Response 7:  Changes have been made accordingly, line 118:

“The genetic population structure was analysed using the program STRUCTURE version 2.3.4 [27]. The admixture model with allele frequencies correlated and the settings 500,000 iterations after a burn-in period of 100,000 were chosen. Twenty runs for K1 to K5 were computed.”

Point 8: Material and methods, lines 170-171: this sentence is awkward, something seems to be missing.

Response 8: Changes have been made accordingly, line 181:

“When applied to simulated populations BIC showed the most accurate results while the marginal R2 had drawbacks due to the bias towards more complex models of the simulation [45].”

Point 9: Results, lines 229-230: Figure 2 is labelled as Figure 1.

Response 9: Figure labelling was corrected. 

Point 10: Results, lines 233-234: I think it's more appropriate to qualify genetic distances as "high" rather than "great". Please check throughout the text.

Response 10: Low, moderate and great genetic differentiation are actually the appropriate terms defined after Wright (1978). However, if you feel like this could be confusing to the readership, we can alter these. Regardless, a reference to Wright (1978) was added to the text, line 236:

“PA showed overall great genetic distance (categorized according to [18]) to all study sites and very great to FL and SH.”

Point 11: Results, lines 247-249: what do you mean "or lack thereof"? how can one population simultaneously concentrate most migration and "lack thereof"? Also, is the rate of self-migration the same as the proportion of individuals classified as residents by BayesAss? At any rate, I would avoid the use of "self-migration", which can be misleading.

Response 11: Changes have been made accordingly to avoid the term self-migration and explain the migration around LI further (line 249):

“Estimated migration rates between study sites using BayesAss3 showed that most migration towards other study sites was found around LI. Except for MB there was almost no migration towards this site, however individuals seem to have migrated from LI into other study sites, mostly WE, BE and SH. PA displayed the lowest levels of migration which further indicates the genetic isolation of this study site.”

Point 12: Results, line 263 (Legend to table 4): some information seems to be missing; the last sentence seems incomplete.

Response 12: Changes have been made accordingly, line 264:

“Model selection analysis with delta Bayesian Information Criterion (“BIC”) and marginal R2 as indicators showing the effects of different landscape structure models on population differentiation (FST) and migration rate (“mig”) for pairs of B. variegata subpopulations. LogLik for complex models was calculated between each model and an undifferentiated landscape (“UNDIF”).”

Point 13: Results, line 279: delete second comma in "Clearly, the models, assigning" (replace by "Clearly, the models assigning...".

Response 13: Changes have been made accordingly (line 282).

Point 14: Discussion, line 300: see comment above on the use of "great".

Response 14: See response 10.

Point 15: Discussion, line 305: "increasing", not "incresaing".

Response 15: Changes have been made accordingly (line 296).

Point 16: Discussion, line 306: see comment above on clarifying the number of individuals genotyped in previous studies: you used the same individuals or more?

Response 16: Yes, this had a wording error in it, we corrected it accordingly, line 308:

“With the addition of the microsatellite B13 to the data set and increasing the number of individuals by combining two datasets [20;22] a total of four genetic clusters were found compared to the previous study with three clusters examining the same study sites [20].”

Point 17: Supp. Material, line 396: What do allelic richness estimates refer to, are they population averages? Please explain

Response 17: Changes have been made accordingly, line 400:

“Summary statistics for the used microsatellite markers (loci) averaged across all study sites. Number of detected alleles, expected heterozygosity (“He”), observed heterozygosity (“Ho”), size range of each loci in base pairs, allelic richness of each loci and confidence interval (“CI”). “Tetra” and “di” refers to the number of repeats (tetranucleotide or dinucleotide). Ten loci in 440 individuals analysed.”

Reviewer 2 Report

This work is to analyses genetic structures of the endangered yellow-bellied toad (Bombina variegata) in the Northern Weser Hills of Germany, and then test impact of different landscape elements on the genetic connectivity of the subpopulations in this area and determ landscape permeability for yellow-bellied toads. Generally speaking, this work are very important for conservation of the endangered yellow-bellied toad. I suggest the authors to improve the materials and methods before acception for publication.

  1. Figure 1. should note what are the different white lines.
  2. Each of the study sites should be clear what kind habitat.
  3. Also need telling us what samples had been collected and how to collect these samples.
  4. Table1. 

    Here the value need being re-estimated and improved.  
    Agricultural Land should be told what planted and cultivation method and cropping system. 
    Rivers & Lakes:different size of them should have different  resistance value  

    In fact, these  Landscape category 's resistance value can be tested   in this work.

  5. Line 225: the four (BE, RO, SH and WE) sites' populations should be introduced in the introduction section that they are reintroduced populations and inform why included them in the study.

Author Response

Dear Reviewer,

Herewith we submit a revised version of our manuscript entitled “Landscape genetics of the Yellow-Bellied Toad (Bombina variegata) in the Northern Weser Hills of Germany”. We thank the referees, reviewers and the editors for their comments that helped to improve the text. We did our best to correct all mistakes listed by the reviewers.

The revised version was not submitted elsewhere and we hope it is acceptable for publication in Diversity in its current form. We adhere to the guidelines for the use of animals in research and the laws of the country where the research was conducted. The manuscript contains only material that is either original or stems from publications identified by a reference. All authors have agreed to be listed, approved the final manuscript and take responsibility for its contents.

Sincerely,

Jasmin Kleißen

Point 1: Figure 1. should note what are the different white lines.

Response 1: Changes have been made accordingly, line 90:

“Figure 1. Study area, locations of the 15 sample sites in the Northern Weser Hills and geographic distribution of the four estimated genetic clusters. Coloration indicates the membership of the individuals to the given cluster. Pie chart size is relative to the number of sampled individuals per study site. White lines indicates the border between federal states and localities. Fuchsloch (“FL”), Edler/Brinkmeyer (“BM”), Bokshorn (“BH”), Wülpker Egge (“WE”), Messingsberg (“MB”), Schlingmühle (“SM”), Bernsen (“BE”), Liekwegen (“LI”), Waldwiese (“WW), JBF-Wiese (“JW”), Borstel (“BO”), Rohden (“RO”), Nato-Station (“NS”), Segelhorst (“SH”) and Pötzen (“PA”). * = n < 10. † = reintroduced populations, n total individuals = 440.”

Point 2: Each of the study sites should be clear what kind habitat.

Response 2: A description of each habitat/study site was added to table 3.

Point 3: Also need telling us what samples had been collected and how to collect these samples.

Response 3: Changes have been made accordingly, line 105:

“The reintroduced sites were excluded for the landscape genetic analysis of this study since they could falsify the results. A total of 440 individuals were analysed for this study (see table 3 for number of individuals tested at each study site). Buccal swab samples were collected from the toads during the breeding season from May to September 2016, as described in [20].”

And line 110:

“Previously published data [20;22] for nine microsatellite markers [23,24] were used for this study in order to obtain a larger dataset for the population and landscape genetic analysis. In [20] the amplification of the microsatellite B13 failed in 2016 but was successfully amplified for this study. Therefore, a total of ten microsatellite loci are collectively analysed in this study.”

Point 4: Table1. Here the value need being re-estimated and improved.  
Agricultural Land should be told what planted and cultivation method and cropping system. 
Rivers & Lakes:different size of them should have different  resistance value.  In fact, these  Landscape category 's resistance value can be tested   in this work.

Response 4: There is not really a way to distinguish between the different crop compositions or identify extensive or intensive use of agricultural land since there is no such data available.

A differing width in the Weser (the river in this area) does not impact the toad’s migration pattern as they would either get drifted away by the water velocity or won´t cross the river due to its depth which increases the predatory pressure imposed by fish. I have consulted experts in the field, they agreed that toads are opposed to cross waters deeper than 50 cm. So, this is not only an issue of width, but more so the water depth. Also, wider parts of the river are actually represented, since the barrier interpretated by Circuitscape is “thicker” in these area, since more pixels are coded with the resistance value 9 (strong barrier). Also, different resistance values were assigned to the Weser, see model WESER6 and WESER3 in table 2, and these were tested, see table 4. A corresponding explanation was added in line 155:

“Due to its depth and velocity, the river “Weser” in this area was considered a strong dispersal barrier [6], regardless of differing width of the water body [42, personal communication].”

Also, the name Rivers & Lakes was changed to Weser, as we believe the other term might have been too broad.

Point 5: Line 225: the four (BE, RO, SH and WE) sites' populations should be introduced in the introduction section that they are reintroduced populations and inform why included them in the study.

Response 5: These study sites are actually excluded from collective calculations (landscape genetics) and are clearly marked in all other calculations. Therefore, it is easy for the reader to identify these sites and interpretate their impact. Throughout the text the fact that some study sites have been reintroduced in the past is mentioned and it is pointed out, that these results should be read within this context. Also, we believe that excluding the reintroduced sites in all further calculations and examinations of the toads in the Northern Weser Hills would rob these studies of important progress-data regarding these specific sites. It would be interesting to see, how these anthropogenic changes are detectable in a couple year or even decades in the future and studies like these are important reference material. 

We added this section to the introduction, line 103:

“Between 2013 and 2016 around 7,000 larvae were released at the study sites BE, Rhoden (“RO”), SH and WE as part of a reintroduction program of the Nature and Biodiversity Conservation Union (“NABU”) Lower Saxony [21]. These individuals stem mostly from LI but also from BM. The reintroduced sites were excluded for the landscape genetic analysis of this study since they could falsify the results. A total of 440 individuals were analysed for this study (see table 3 for number of individuals tested at each study site). Buccal swab samples were collected from the toads during the breeding season from May to September 2016, as described in [20].”
